# Immune Landscape and Application of Immune Checkpoint Inhibitors in Clear Cell Renal Cell Carcinoma

**DOI:** 10.3390/ijms262411986

**Published:** 2025-12-12

**Authors:** Yanhe An, Na Luo

**Affiliations:** Department of Anatomy and Histology, School of Medicine, Nankai University, Tianjin 300071, China; anyanhe1214@163.com

**Keywords:** clear cell renal cell carcinoma, immune checkpoint inhibitors, targeting therapy

## Abstract

Clear cell renal cell carcinoma (ccRCC) represents the predominant histological subtype of renal cell carcinoma (RCC), constituting approximately 85% of all RCC cases. Recent advancements in therapies aimed at targeting angiogenesis have marked a significant breakthrough in the treatment of ccRCC, with several of these therapies receiving approval for clinical use. Furthermore, the introduction of immune checkpoint inhibitors (ICIs) has demonstrated efficacy in the management of ccRCC. Nonetheless, there is an urgent need for the identification of predictive and prognostic biomarkers, which are currently under investigation. This review offers an extensive examination of the pathological, genomic, and molecular characteristics of ccRCC, with particular emphasis on its immune attributes. Additionally, it addresses the clinical implications of targeted therapies and immunotherapies, whether administered as monotherapy or in combination with traditional or novel agents, while also evaluating the results of pertinent clinical trials. By encompassing a wide range of topics related to ccRCC, from foundational knowledge to clinical applications, this review aims to deepen the understanding of the essential features of ccRCC and to establish a theoretical basis for the formulation of clinical strategies.

## 1. Introduction

Kidney cancer is among the 10 most common cancers in both men and women. The lifetime risk for developing renal cell cancer is about 2% in men and 1% in women [1]. There were 434,419 new cases and 155,702 deaths worldwide due to kidney cancer in 2024 according to the Global Cancer Observatory [2].

Clear cell renal cell carcinoma (ccRCC) is the most common histological type of renal cell carcinoma (RCC). ccRCC is a heterogeneous group of cancers arising from renal tubular epithelial cells and accounts for about 85% of all RCC cases [3]. Unfortunately, about 30% of RCC patients present with a distant metastasis at initial diagnosis. And, the 5-year relative survival rate of RCC patients is approximately 80%, of which metastatic RCC (mRCC) accounts for about 15% [4]. Surgery is the preferred option for localized RCC, although the exploration for more effective treatments remains ongoing. Moreover, RCC is generally regarded as resistant to both radiation therapy and chemotherapy. Nevertheless, stereotactic body radiotherapy (SBRT) has demonstrated efficacy in controlling the progression of primary or metastatic RCC in cases characterized by oligoprogressive or oligometastatic disease, thereby prolonging the duration of systemic therapy [5]. Consequently, SBRT is increasingly recognized as a viable alternative treatment modality for patients with RCC, either as a standalone intervention or in conjunction with other therapeutic approaches, irrespective of ccRCC or non ccRCC histology [6]. High-dose interleukin 2 (IL-2) and interferon-α (IFN-α) cytokine therapies have been established as primary treatment modalities for patients with mRCC. However, the associated toxicity and adverse effects significantly restrict their clinical applicability [7,8]. The introduction of targeted therapy has provided new treatment alternatives for numerous patients with mRCC, particularly following the approval of Sorafenib for this indication in 2005 [9].

Furthermore, the emergence of immune checkpoint inhibitors (ICIs) has demonstrated clinical benefits in the first-line treatment. In this review, we summarize the pathological, genomic/molecular, and especially the immune landscape of ccRCC. We hope that understanding the specific features of ccRCC discussed in this review will stimulate new ideas that contribute to the development of novel treatments for ccRCC.

## 2. Characterization of ccRCC

### 2.1. Pathological Characterization of ccRCC

Typically, ccRCC appears as epithelial cells with a transparent cytoplasm and a well-defined cell membrane. The transparent cytoplasm results from an accumulation of glycogen, phospholipids, and neutral lipids [10,11]. The various ccRCC patterns include a compact-alveolar (nested), tubular (acinar), or a microcystic pattern. The compact-alveolar (nested) pattern consists of rounded epithelial nests separated by a highly vascularized connective tissue that results in a sinusoidal appearance. The tubular (acinar) pattern consists of epithelial acini with central lumens separated by a vascularized connective tissue. In addition, in regions of cystic degeneration, the acini dilate to form a microcystic and/or macrocystic pattern with the lumens containing necrotic material, pale eosinophilic fluid, and red blood cells. ccRCCs often contain more than one of the above-mentioned patterns [11].

### 2.2. Genomic/Molecular Characterization of ccRCC

Most ccRCC patients demonstrate a chromosome 3p loss that leads to the loss of multiple tumor-suppressor genes and subsequently to ccRCC tumorigenesis. In addition to the chromosome 3p loss, the most common mutated genes involved in ccRCC include: ① von Hippel–Lindau Tumor Suppressor (VHL), ② Polybromo 1 (PBRM1), ③ SET Domain Containing 2 (SETD2)-Histone Lysine Methyltransferase, ④ Lysine Demethylase 5C (KDM5C), ⑤ BRCA1 Associated Protein 1 (BAP1), ⑥ Phosphatase And Tensin Homolog (PTEN), ⑦ Mechanistic Target Of Rapamycin Kinase (mTOR), ⑧ Tumor Protein P53 (TP53), and ⑨ Phosphatidylinositol-4, 5-Bisphosphate 3-Kinase Catalytic Subunit Alpha (PIK3CA) [12,13,14].

The von Hippel–Lindau tumor suppressor (VHL) gene is the most mutated gene in ccRCC involving 80–90% of ccRCC cases, which suggests that the loss of VHL function is the key driver in ccRCC. VHL belongs to the substrate recognition component of the E3 ligase complex that drives degradation of Hypoxia Inducible Factors alpha (HIFα). A VHL mutation stabilizes HIF-1/2-α and induces the expression of several angiogenesis-related factors [e.g., vascular endothelial growth factor (VEGF), platelet-derived growth factor (PDGF), and transforming growth factor-β (TGF-β)], all of which play important roles in tumorigenesis of ccRCC [15,16,17]. The PI3K/AKT signaling pathway also frequently undergoes genetic alterations in ccRCC which include PTEN deletions or mutations, mTOR mutations, and PIK3CA amplifications or mutations. The resultant activation of the PI3K/AKT signaling pathway contributes to ccRCC initiation or progression [18]. In addition, numerous epigenomic-related gene mutations occur in ccRCC (e.g., PBRM1, SETD2, KDM5C, BAP1 deletions or mutations) which suggests that dysregulation of chromatin remodeling plays an important role in ccRCC [18]. In a Chinese ccRCC cohort, the most common genomic alterations occur in the VHL, TP53, SETD2, BAP1, and PBRM1 genes with VHL and TP53 genes demonstrating a high mutation frequency and the PBRM1 gene demonstrating a low mutation frequency [19]. Furthermore, kidney injury molecule-1 (KIM-1), a transmembrane protein that is overexpressed in clear cell renal cell carcinoma (ccRCC), has been recognized as both a diagnostic and prognostic biomarker [20]. The expression of KIM-1 has been demonstrated to correlate with immune cell infiltration and the efficacy of immunotherapeutic responses in ccRCC [20,21].

### 2.3. Immune Landscape of ccRCC

ccRCC shows a distinct immune characterization when compared to other cancers. Generally, a cancer with a high tumor mutation burden (TMB) correlates with an increase in tumor-infiltrating lymphocytes (especially cytotoxic CD8^+^ T cells) and human leukocyte antigen (HLA) heterozygosity, which portend a good outcome in response to immunotherapy. However, ccRCC behaves in an unusual way in that ccRCC with a low TMB shows a sensitivity to immunotherapy. Moreover, ccRCC behaves in another unusual way in that ccRCC with an increased infiltration of cytotoxic CD8^+^ T cells portends a poor outcome in response to immunotherapy, instead of a good outcome. And, HLA heterozygosity does not play a role in ccRCC [22].

Despite its unusual immune characteristics, ccRCC is recognized as an immunogenic tumor characterized by significant heterogeneity in its immune phenotypes. Various analytical approaches, including genomic, transcriptomic, proteomic, metabolomic, and spatial transcriptomic analyses, have been employed to elucidate the tumor immune microenvironment associated with ccRCC. While techniques such as flow cytometry, mass cytometry, and multiplexed immunohistochemistry have been utilized to identify cell types within this microenvironment, these methods are constrained by the limited availability of antibodies, which restricts the identification to only those cell types that are anticipated. Consequently, bulk RNA sequencing transcriptomic analysis, particularly single-cell RNA sequencing, has emerged as the primary methodology for assessing the immune cell composition in ccRCC [23].

Recent single-cell RNA sequencing studies have elucidated the pivotal roles of macrophages and T cells within the tumor microenvironment, particularly in ccRCC [22,24]. Notably, ccRCC exhibits a heightened infiltration of both CD8^+^ T cells and macrophages when compared to normal renal tissue. The CD8^+^ T cells demonstrate a transcriptional continuum that ranges from a naïve state to activation, ultimately leading to a state of exhaustion characterized by highly expanded clonotypes [23]. As ccRCC advances from early to late stages, there is a discernible increase in the population of terminally exhausted CD8^+^ T cells, accompanied by a concomitant decrease in cytotoxic CD8^+^ T cells. The exhausted T cells observed in advanced stages exhibit a phenotype indicative of metabolic insufficiency, characterized by dysfunctional mitochondria and impaired glucose uptake. The gene PPARGC1A has been identified as the most significantly down-regulated gene in advanced disease, correlating with mitochondrial dysfunction in T cells [25]. Sumeyye et al. further reported that the predominant immune cell types identified in ccRCC include CD8^+^ T cells, macrophages, and CD4^+^ T cells [26]. Additionally, ccRCC can be stratified into four distinct groups based on the relative abundance of immune cells. The first group is characterized by a comparable number of CD8^+^ T cells and macrophages, with a lower presence of CD4^+^ T cells. The second group exhibits a markedly higher number of macrophages relative to other immune cell types. The third group is distinguished by a significantly elevated number of CD8^+^ T cells compared to other immune cells. Finally, the fourth group contains a similar number of CD4^+^ T cells and macrophages, but with fewer CD8^+^ T cells. In addition, the proportion of mast cells has been shown to correlate with the progression of ccRCC in some of the aforementioned groups [26].

Furthermore, Borcherding et al. reported the existence of eight distinct clusters of CD8^+^ T cells in ccRCC using single-cell RNA sequencing and T-cell-receptor sequencing. Some clusters showed increased expression of immune checkpoint inhibitors (e.g., CTLA4, HAVCR2, PDCD1, and TIGIT) while other clusters showed increased expression of cytolytic genes (e.g., PRF1 and IFNG) [22]. Moreover, nine distinct clusters of CD4^+^ T cells exist in ccRCC, whereby each cluster shows a differential expression of certain genes involved in various signaling pathways. Some clusters demonstrate a high level of cytolytic and type I signaling while other clusters demonstrate a high expression of terminal differentiation markers.

In their investigation of the immune characteristics of metastatic sites in ccRCC, Kippenberger et al. utilized immunohistochemical staining to compare ccRCC specimens with pancreatic metastases to those without such metastases. Their findings revealed a notable decrease in the intratumoral infiltration of CD8^+^ T cells and immunosuppressive FOXP3^+^ T cells within the pancreatic metastases when contrasted with the corresponding primary tumors [27]

Borcherding et al. identified five distinct clusters of macrophages in ccRCC, each exhibiting a unique expression profile. Notably, none of these macrophage clusters corresponded to the M1 or M2 polarization states. It is posited that further investigations aimed at the detailed characterization and comprehension of immune cells with specific expression patterns in ccRCC could enhance the prediction of clinical outcomes and facilitate the development of therapeutic strategies for this malignancy [22]. In a study utilizing Vhl wild-type (WT) and Vhl knockout (KO) murine models of kidney cancer, Wolf et al. observed that Vhl KO tumors exhibited a higher infiltration of immune cells. The macrophages within Vhl KO tumors demonstrated increased glucose consumption and inflammatory transcriptional profiles, while T cells in these tumors displayed diminished activation and responsiveness to anti-PD-1 therapy [28].

Yang et al. employed single-cell RNA sequencing (scRNA-seq) and flow cytometry to characterize B cells in ccRCC, identifying three subtypes of pro-metastatic B cells based on extensive genetic profiling: MARCH3, B2M, and DTWD1. Among these, the B2M-B cell subtype was found to be the most critical in facilitating distant metastasis. Additionally, plasma cells in ccRCC were categorized into two subpopulations characterized by the expression of RPS12 and IGHG4 genes, respectively. The ratio of RPS12^+^ plasma cells to IGHG4^+^ plasma cells was found to be less than one in control tissues, whereas it exceeded one in ccRCC tissues [29].

Moreover, beyond the metabolic conditions and VHL status associated with ccRCC, N6-methyladenosine (m6A) modification has been shown to impact the TME in ccRCC. The m6A score has been identified as an independent prognostic factor for ccRCC, with increased Th2-cell infiltration correlating with high expression levels of IGF2BP3, an m6A reader, which is associated with poor survival outcomes in ccRCC patients [30].

## 3. Development of Therapeutic Treatment for ccRCC

### 3.1. Targeting Therapy

Since VHL mutation induces the expression of several angiogenesis-related factors in ccRCC, targeting therapies directed at angiogenesis have been developed and approved for ccRCC treatment. These ccRCC targeting therapies include tyrosine kinase inhibitors (TKIs) [e.g., sunitinib, pazopanib, sorafenib, axitinib, cabozantinib, lenvatinib, and bevacizumab] and inhibitors of the mTOR signaling pathway (e.g., everolimus, temsirolimus) (Table 1, Figure 1).

Sorafenib is a multi-kinase inhibitor that primarily inhibits VEGFR1, 2, 3, and PDGF receptor (PDGFR). Sorafenib also has an action on newly identified targets CRAF, BRAF, and V600E BRAF. Sorafenib is the first commercialized TKI for metastatic ccRCC and European Medicines Agency (EMA) approved second-line treatment of metastatic ccRCC that was resistant to IL-2 or IFN-α therapy [31,32,33].Sunitinib is a multi-kinase inhibitor that primarily inhibits VEGFR1, 2, 3, and PDGFR. Due to a clinical trial in 2007, which shows a better overall response rate (ORR) and a longer progression-free survival (PFS) of sunitinib group compared to IFN-α group, sunitinib/targeted therapy replaced cytokine therapy becoming the standard care for metastatic ccRCC. Sunitinib is the Food and Drug Administration (FDA)- and EMA-approved first-line treatment of metastatic ccRCC [34,35,36,37].Pazopanib is a multi-kinase inhibitor that primarily inhibits VEGFR1, 2, 3, and PDGFR. It shows a better safety profile without compromising the efficacy compared to sunitinib. Pazopanib is the EMA-approved first-line treatment of metastatic ccRCC. Sunitinib and pazopanib are the favored targeted therapies for the first-line treatment of ccRCC [38,39].Axitinib is a second-generation of TKI that selectively targets VEGFR 1, 2, and 3, whose IC50 is of picomolar level and significantly lower than the first-generation of TKIs. Axitinib is an FDA- and EMA-approved second-line treatment of metastatic ccRCC that was resistant to sunitinib or cytokine treatment [40,41,42,43].Cabozantinib is a multi-kinase inhibitor that targets VEGFR, mesenchymal epithelial transition factor (MET), and anexelekto (AXL), all of which are commonly upregulated in RCC. Cabozantinib is an FDA- and EMA-approved first-line treatment of metastatic ccRCC with intermediate or poor risk and second-line treatment of metastatic ccRCC that was resistant to anti-VEGF therapy [44,45,46,47,48,49,50,51,52,53,54].Lenvatinib is a multi-kinase inhibitor that primarily inhibits VEGFR1, 2, 3, fibroblast growth receptor (FGFR)1, 2, 3, 4, and PDGFRα. Lenvatinib is an FDA-approved first-line treatment of metastatic ccRCC, and generally used in combination with other drugs [55,56,57,58,59,60,61].Tivozanib is a tyrosine kinase inhibitor that is selectively targets VEGFR1, 2, and 3, which works at picomolar level to inhibit VEGFR phosphorylation. It has a better safety profile than the broad-spectrum tyrosine kinase inhibitors and is recommended as a third- or fourth-line treatment for metastatic ccRCC [62].Bevacizumab is a recombinant humanized anti-VEGF monoclonal antibody that completely binds circulatory VEGF and prevents VEGFR activation. However, it is not used in the clinical treatment for ccRCC [63,64,65,66,67,68,69].Everolimus is a derivative of Rapamycin that binds FK506 binding protein 12 (FKBP12) and inhibits mTOR activity resulting in a G1 growth arrest and decreased levels of both HIFs and VEGF. Everolimus is an oral inhibitor and an EMA-approved first-line treatment of metastatic ccRCC and a recommended drug after the failure of first-line TKI therapy [70,71,72].Temsirolimus binds FK506 binding protein 12 (FKBP12) and inhibits mTOR activity resulting in a G1 growth arrest and decreased levels of both HIFs and VEGF. Temsirolimus is administered intravenously. FDA and EMA approved temsirolimus as a first-line treatment of metastatic ccRCC. mTOR inhibitors are more potent in inhibiting cell proliferation than neovascularization [73,74,75].

The prevalent side effects associated with TKIs include hypertension, hand-foot syndrome, vomiting, and diarrhea, alongside with other symptoms such as fatigue, reduced appetite, nausea, hypothyroidism, and stomatitis [72]. Despite the notable enhancement in ORR, overall survival (OS), and PFS observed in specific patients with ccRCC following TKI treatment, the issues of drug resistance and hyposensitivity continue to pose significant challenges in the management of ccRCC.

**Table 1 ijms-26-11986-t001:** Inhibitors of targeting therapy for ccRCC.

Inhibitors	Targets	Mechanisms	Clinical Application
Sorafenib [31,32,33]	VEGFR-1/-2/-3, PDGFR-α/-β, BRAF V600E, c-Raf, c-kit, FLT-3	blocking RAF/MEK/ERK signaling pathway and suppressing VEGFR and PDGFR	RCC, hepatocellular carcinoma, breast carcinoma, colorectal carcinoma, thyroid cancer, myeloid leukemia
Sunitinib [34,35,36,37]	VEGFR-1/-2/-3, PDGFR-α/-β, FLT-3, c-kit	suppressing VEGFR-2 signaling	advanced ccRCC, gastrointestinal stromal tumors, breast cancer, small cell lung cancer
Pazopanib [38,39]	VEGFR-1/-2/-3, PDGFR-α/-β, LCK, c-fms, FGFR-1/-3, c-kit	inhibiting VEGFR-2 phosphorylation	advanced RCC, soft tissue sarcoma
Axitinib [40,41,42,43]	VEGFR-1/-2/-3, PDGFR, KIT	selectively inhibiting VEGFR-1/-2/-3 signaling	advanced RCC, advanced sarcoma, head and neck malignancies, advanced non-small-cell lung cancer
Cabozantinib [44,45,46,47,48,49,50,51,52,53,54]	VEGFR-2, MET, RET, AXL, FLT3, c-kit	targeting MET and VEGFR2	advanced RCC, advanced hepatocellular carcinoma, progressive/metastatic medullary thyroid cancer, adrenocortical carcinoma, breast cancer, glioblastoma, non-small cell lung cancer, melanoma, ovarian cancer
Lenvatinib [55,56,57,58,59,60,61]	VEGFR-1/-2/-3, PDGFR, c-kit, FGFR-1/-2/-3/-4	dual inhibition of VEGFR1-3 and FGFR1-4 signaling	thyroid carcinoma, unresectable hepatocellular carcinoma, unresectable thymic carcinoma, advanced RCC,
Tivozanib [62]	VEGFR-1/-2/-3, c-kit, PDGFR-β	selectively inhibition of VEGFR-1/-2/-3	advanced RCC
Bevacizumab [63,64,65,66,67,68,69]	VEGF	preventing the activation of VEGFR1 and VEGFR2 on endothelial cells	metastatic colorectal cancer, advanced non-small cell lung cancer, metastatic RCC, glioblastoma, advanced cervical cancer, ovarian cancer
Everolimus [70,71,72]	mTOR complex 1	inhibiting mTOR activity, which results in a G1 growth arrest, and decreased levels of HIFs and VEGF	Pancreatic, gastrointestinal, and pulmonary neuroendocrine tumors, RCC, hormone receptor-positive, HER2-negative breast cancer
Temsirolimus [73,74,75]	mTOR complex 1	inhibiting mTOR activity, which results in a G1 growth arrest, and decreased levels of HIFs and VEGF	advanced RCC, mantle cell lymphoma, radio-resistant nasopharyngeal carcinoma

### 3.2. Immunotherapy

◆Cytokine

IL-2 and recombinant human IFN- α are the first approved immunotherapies for ccRCC. However, a low overall objective response rate and a high toxicity to multiple organs limits the application of cytokine therapy [8].

◆Immune Checkpoint Inhibitors (ICIs)

Immune checkpoints refer to the gatekeepers of immune response that maintain immune tolerance and suppress autoimmunity [76]. Cancer cells express immune checkpoints on their cell membrane that give cancer cells the ability to evade immune surveillance so that cancer progression can continue. Thus, immune checkpoint inhibitors (ICIs) that block the cancer cell immune checkpoint ability to evade immune surveillance have been developed and have achieved unprecedented clinical success [77]. Currently, the FDA has approved a variety of ICIs for cancer treatment. For example, ICIs that target cytotoxic T lymphocyte antigen 4 (CTLA-4), programmed cell death 1 (PD-1), and programmed cell death ligand 1 (PD-L1) have been developed. In addition, ICIs that target Lymphocyte activating gene-3 (LAG3), T-cell Immunoreceptor with immunoglobulin and ITIM domain (TIGIT), and T-cell Immunoglobulin and mucin domain-containing 3 (TIM3) have also been widely investigated. Furthermore, ICIs that target V-domain immunoglobulin suppressor of T cell activation (VISTA), indoleamine 2,3-dioxygenase 1 (IDO1) and triggering receptor expressed on myeloid cells-2 (TRME2) are undergoing investigation for the treatment of ccRCC [78,79,80] (Table 2, Figure 2).

**CTLA-4** expression occurs in a variety of cancers and associates with T cell infiltration [81]. CTLA-4 is homologous to CD28 and expressed on T cells. CTLA-4 competitively recruits CD80/CD86 and forms higher affinity than CD28 that limits its interaction with CD28 and plays an inhibitory role in the regulation of T cell activation [82,83].**PD-1** expression occurs mainly in activated T cells, B cells, and natural killer (NK) cells with an elevated expression in tumor-specific T cells [84]. **PD-L1** (a PD-1 ligand) expression occurs on the cell membrane of tumor cells. In general, PD-1 expression decreases as the antigen decreases. However, PD-1 expression increases and leads to T cell exhaustion if the antigen is present for a long time [85]. PD-1 binding to PD-L1 prevents T cell activation and proliferation via inhibition of the PI3K-AKT-mTOR and Ras-EMK-ERK signaling pathways, which thereby attenuates the killing action of T cells [86]. If PD-1 binding to PD-L1 is blocked, then the state of T cell exhaustion can be restored [87]. PD-L1 and PD-L2 are highly expressed in both primary and metastatic sites in advanced ccRCC [87,88].**LAG3** (an inhibitory receptor) expression occurs mainly in activated T cells, Tregs, and NK cells [89]. LAG3 binds MHC-II with a high affinity and prevents MHC-II interaction with T cell co-receptor CD4 [90,91]. Dual blockade of LAG3 and PD-L1 results in synergistic anti-tumor effects since LAG3 and PD-L1 co-expression always occurs in tumor-infiltrating T cells [92]. Furthermore, LAG3 and PD-1 combinations are common in activated T cells in ccRCC tissues. Consequently, dual blockage of LAG3 and PD-1 holds promise as an effective ccRCC treatment [93].**TIGIT** (an inhibitory receptor) expression occurs mainly in activated CD8^+^ or CD4^+^ T cells, Tregs, and NK cells [94]. CD155, CD112, and CD113 (TIGIT ligands) expression occurs in APCs. CD155 demonstrates the highest affinity for TIGIT among these ligands. TIGIT binding to ligand alters APC function and reduces cytokine release which results in diminished T cell activation [95]. Studies have shown that TIGIT is expressed on exhausted CD8^+^ T cells, and detected in peripheral blood mononuclear cells, and TILs in ccRCC patients [96].**TIM3** (a co-inhibitory receptor) expression occurs mainly in IFN-γ-producing T cells, FOXP3^+^ Treg cells, and innate immune cells [97]. Galectin-9 (TIM3 ligand) is found on the surface of cancer cells or in the parenchyma. TIM3 binding to galectin-9 results in disruption of immune synapse formation and ultimately anergia or apoptosis of T cells. The increase in TIM3 expression results in a suppression of the T cell response and T cell dysfunction, similar to PD-1 [98]. Tumor-infiltrating lymphocytes (TILs) that express TIM3 also express PD-1 and TIM3^+^PD-1^+^ TILs exhibit a more severe depletion phenotype [99]. Consistent with the above findings, TIM3 and PD1 co-expression on CD8^+^ T cells results in poor clinical outcomes for ccRCC patients [100].**VISTA** expression occurs in a variety of tumor cells and TILs. It shares a homology with PD-L1 and plays a role in tumor immunosuppression [101]. VISTA serves dual immunosuppressive roles as both a ligand on APCs with PSGL-1 being its receptor on CTLs and a receptor on CTLs with VSIG-3 as its ligand [102]. VISTA interacts with its receptor/ligand, resulting in the inhibition of T cell activation and proliferation, while simultaneously promoting the expression of Foxp3 within the TME [103]. VISTA expression occurs mainly in CD14^+^HLA^−^DR^+^ macrophages and activated CD8^+^ T cells [104,105], and is significantly increased in ccRCC, even more than PD-L1.**IDO1** is an amino acid cytosolic haem-containing enzyme involved in the first, rate-limiting step of the tryptophan metabolism to kynurenine. It is associated with increased introtumoral Treg infiltration and impaired cytotoxic T-cell function. IDO1 plays an immunosuppressive role illustrated in two aspects. On one hand, local depletion of Trp results in the activation of the amino-acid-sensitive GCN2 and mTOR stress-kinase pathways, which in turn causes cell cycle arrest and induction of anergy of responding T cells. On the other hand, the downstream Kyns induces effector T-cell arrest or apoptosis and may contribute to the conversion of naive CD4+ T cells into FOXP3+ Treg cells [106]. Moreover, it also contributes to MDSC infiltration and M2 polarization [107].**TREM2** expression occurs in tumor-associated macrophages (TAMs) which are immunosuppressive cells in the TME [105]. TAMs cause T cell dysfunction, tolerance to PD1/PD-L1 therapy, and poor clinical outcomes [106]. TAMs that express a high level of TRME2 are more abundant in ccRCC than in normal renal tissue [107]. In this regard, high TRME-2-expressing TAMs are often accompanied by a higher risk of recurrence. Therefore, TRME-2 is a putative therapeutic target for the treatment of ccRCC, and clinical trials using TRME2 blockade therapy are currently ongoing [108].

**Table 2 ijms-26-11986-t002:** Current and emerging inhibitors of immune checkpoint for ccRCC.

Targets	Molecular Properties	Mechanism	Inhibitors
CTLA-4	expressed on T cells, structurally homologous to CD28 and competitively binds CD80/CD86	CTLA-4 competes with CD28 for CD80/CD86 binding and acting as an antagonist of CD28-mediated co-stimulation of T cells [81,82,83]	Ipilimumab [108]
PD-1	expression occurs mainly in activated T cells, B cells, and NK cells, and binds PD-L1 and PD-L2	PD-1 binding to PD-L1 prevents T cell activation and proliferation via inhibition of the PI3K-AKT-mTOR and Ras-EMK-ERK signaling pathways [86,87]	Nivolumab [109,110,111] Pembrolizumab [112]
PD-L1	expression occurs in tumor cells and binds to PD-1	PD-1 binding to PD-L1 prevents T cell activation and proliferation via inhibition of the PI3K-AKT-mTOR and Ras-EMK-ERK signaling pathways [86,87]	Avelumab [113] Atezolizumab [114]
LAG3	expression occurs mainly in activated T cells, Tregs, and NK cells	LAG3-MHC-II interface overlaps with the MHC-II-binding site of the CD4, disrupting CD4-MHC-II interactions as a mechanism for LAG3 immunosuppressive function [91]	Relatlimab (NCT02996110) [115] Ieramilimab (NCT05148546) [115]
TIGIT	expressed on activated CD8+ or CD4+ T cells, Tregs, and NK cells, and binds CD155, CD112 and CD113 on APC	TIGIT binding to CD155 alters APC function and reduces cytokine release which results in diminished T cell activation [95]	Tiragolumab (NCT03977467) [115]
TREM2	occurs in tumor-associated macrophages (TAMs)	TREM2 functions as a tumor suppressor and is predominantly expressed in various myeloid cell types, including dendritic cells (DCs), immunosuppressive macrophages, and monocytes. The absence of TREM2 facilitates the phenotypic transformation of macrophages towards the M1 phenotype, accompanied by the increased secretion of pro-inflammatory cytokines such as IL-12, IL-6, IL-15, and TNF. Notably, TREM2 deficiency also enhances the proliferation and activation of both CD8^+^ and CD4^+^ T cells [80]	PY314 (NCT04691375) [115]
TIM3	expressed on IFN-γ-producing T cells, FOXP3+ Treg cells, and innate immune cells, and binds to galectin-9	binding to galectin-9 results in disruption of immune synapse formation and ultimately anergia or apoptosis of T cells [97]	Cobolimab [116] Sabatolimab [117]
VISTA	expression occurs in a variety of tumor cells and TILs, sharing homology to PD-L1	VISTA functions as both a ligand on APCs with PSGL-1 being its receptor on CTLs and a receptor on CTLs with VSIG-3 as its ligand, which potently suppresses activation of T cells [102,103]	HMBD-002 [118] KVA12123 [119]
IDO1	an amino acid cytosolic haem-containing enzyme involved in the first, rate-limiting step of the tryptophan metabolism to kynurenine	1. Local depletion of Trp results in activation of the amino-acid-sensitive GCN2 and mTOR stress-kinase pathways, which in turn causes cell cycle arrest and induction of anergy of responding T cells. 2. Downstream Kyns induces effector T-cell arrest or apoptosis, and may also contribute to the conversion of naive CD4^+^ T cells into FOXP3^+^ Treg cells [106]	Epacadostat [120] Navoximod [121]

◆Application of ICIs monotherapy in ccRCC (Table 3)

■CTLA-4 inhibitor

○Ipilimumab (a CTLA-4 blocker) is used to treat metastatic ccRCC patients and its efficacy was assessed in a phase II clinical trial (NCT00057889). The results based on the Response Evaluation Criteria in Solid Tumors (RECIST) showed the following: 1) 1 of 21 low dose ipilimumab patients had a partial response, 2) 5 of 40 high dose ipilimumab patients had partial responses, and 3) 33% of patients experienced a grade III or IV immune-mediated toxicity. Despite the above-mentioned low overall efficacy of ipilimumab, ipilimumab treatment does induce cancer regression in some metastatic ccRCC patients that were resistant to IL-2 treatment [108].

■PD-1 inhibitor

○Nivolumab shows anti-tumor activity and improves the overall survival (OS) of a variety of malignant tumors, including mRCC [111]. The results based on CheckMate 025 phase III clinical trial (NCT01668784) in 2015 led to FDA approval of nivolumab for the treatment of mRCC patients [110]. In this clinical trial, a total of 821 patients who had received sorafenib or sunitinib were randomly assigned to either nivolumab or everolimus (a mTOR inhibitor) treatment. Of a 72-month median follow-up time, the results showed an ORR of 23% vs. 4%, OS of 25.8 months vs. 19.7 months, and a 5-year survival rate of 23% vs. 4% in the nivolumab versus everolimus groups, respectively. In addition, the results showed an adverse events (AEs) incidence of 80.5% vs. 88.9% and grade 3–4 treatment-related AEs of 21.4% vs. 36.8% in the nivolumab versus everolimus groups, respectively. Thus, nivolumab shows significant survival and safety advantages over everolimus as a second-line treatment for metastatic ccRCC [109].

○Pembrolizumab has shown promise as a first-line treatment in various types of cancer. Pembrolizumab was evaluated as a monotherapy for treating naive patients with metastatic ccRCC in a phase II KEYNOTE-427 clinical trial. In this clinical trial, 110 patients received 200 mg of pembrolizumab intravenously every 3 weeks for 24 months. The results showed a median duration of response of 18.9 months (range; 2–37 months) and a median progression-free survival (PFS) of 7.1 months (95% CI; 5–11 months). The 12-month OS and 24-month OS rates were 88.2% vs. 70.8%, respectively. And, 30% of the patients showed grade 3–5 treatment-related AEs, with colitis (5.5%) and diarrhea (3.6%) as the most common AEs [112].

■PD-L1 inhibitor

○Atezolizumab was evaluated as a first-line treatment for mRCC in the phase II IMmotion150 clinical trial (NCT01984242) [114]. In this clinical trial, one group of patients received 1200 mg of atezolizumab intravenously once every 3 weeks and the other group of patients received 50 mg of sunitinib every day for 4 weeks. The results showed a PFS of 7.8 months vs. 5.5 months, an overall ORR of 25% vs. 29% in the atezolizumab versus sunitinib groups, respectively. Although the overall ORR in the atezolizumab group was lower, the proportion of patients with complete response in the atezolizumab group was higher [114]. In addition, the incidence of treatment-related AEs in the atezolizumab group was less than that in the sunitinib group.

◆Application of ICIs combination therapy in ccRCC (Table 3)

■ipilimumab + nivolumab

Ipilimumab + nivolumab treatment is the only approved dual ICIs combination immunotherapy for low- and intermediate- risk mRCC patients who never received nor currently under any treatments based upon a phase III CheckMate-214 clinical trial [122]. In this clinical trial, 1096 RCC patients with a moderate or poor prognosis received either ipilimumab + nivolumab or sunitinib treatment with a 4-year follow-up [123]. The results showed that the OS and ORR both significantly improved in the ipilimumab + nivolumab group versus the sunitinib group. In addition, patients in the ipilimumab + nivolumab group were more likely to achieve both complete remission or remissions lasting more than 4 years versus the patients in the sunitinib group [124]. The results also showed treatment-related AEs of 94% vs. 97.4% and grade 3–4 AEs of 47.9% vs. 64.1% in ipilimumab + nivolumab versus sunitinib groups, respectively. In an intention-to-treat analysis, the results showed an OS of 55.7 months vs. 38.4 months in ipilimumab + nivolumab versus sunitinib groups, respectively [124,125].

■Emerging + Traditional immune checkpoint inhibitors

With the emergence of many new immune checkpoints, blockade therapy using a combination of both emerging and traditional immune checkpoints has been studied for the treatment of advanced ccRCC. In a phase I clinical trial, patients received 600 mg or 1000 mg of navoximod (an orally active IDO1 inhibitor) twice daily and 1200 mg of atezolizumab once every 3 weeks [121]. The results showed that this combination therapy plays a beneficial role at all dose levels in various tumor types (including RCC) with the most common AEs of fatigue, rash, and pigmentation [121]. However, the full range of clinical advantages and disadvantages of the above-mentioned combination therapies remains uncertain due to the limited number of patients in the clinical trials.

■ICIs combination with anti-VEGF therapy

A variety of ICIs and anti-VEGF combination therapies have been approved for first-line treatment of advanced ccRCC based upon 5 major clinical trials (JAVELIN Renal 101, KEYNOTE-426, KEYNOTE-581, CheckMate9ER, and IMmotion151).

○The phase III JAVELIN Renal 101 (NCT02684006) clinical trial evaluated avelumab (a PD-L1 inhibitor) + axitinib combination therapy compared to sunitinib therapy alone and was the first clinical trial to report on ICI + anti-VEGF combination therapy for RCC [113]. In this clinical trial, one group of 442 PD-L1^+^ RCC patients received avelumab + axitinib combination therapy and the other group of 444 PD-L1^+^ RCC patients received sunitinib therapy. The results showed an ORR of 55.2% vs. 25.5% and a PFS of 13.8 months vs. 8.4 months in the avelumab + axitinib combination therapy group versus the sunitinib therapy group, respectively. The follow-up results (August 2020) showed a PFS of 13.8 months vs. 7.0 months in the avelumab + axitinib combination therapy group versus the sunitinib therapy group, respectively [126,127]. Based upon this clinical trial, avelumab + axitinib combination therapy achieves short-term positive clinical benefits. The OS rate in this clinical trial is currently too early to calculate.

○The phase III KEYNOTE-426 (NCT02853331) clinical trial evaluated pembrolizumab + axitinib combination therapy compared to sunitinib therapy alone. In this clinical trial of 861 ccRCC patients, one group of patients received pembrolizumab + axitinib combination therapy and the other group of patients received sunitinib therapy. The results showed a PFS of 15.1 months vs. 11.1 months and an ORR of 59.3% vs. 35.7% in the pembrolizumab + axitinib combination therapy group versus the sunitinib therapy group, respectively. The 30.6-month follow-up results (October 2020) showed an OS of not reached (NR) vs. 35.7 months and a PFS of 15.4 months vs. 11.1 months in the pembrolizumab + axitinib combination therapy group versus the sunitinib therapy group, respectively [128]. Moreover, the extended follow-up results showed an OS of 57.7% vs. 48.5%, a PFS of 25.1% vs. 10.6%, and an ORR of 60.4% vs. 39.6% in the pembrolizumab + axitinib combination therapy group versus the sunitinib therapy group, respectively [129]. The FDA approved pembrolizumab + axitinib combination therapy as a first-line treatment of advanced ccRCC based upon the above-mentioned results.

○The phase III KEYNOTE-581 (NCT02811861) clinical trial evaluated pembrolizumab + lenvatinib combination therapy compared to sunitinib therapy alone [59,130]. In this clinical trial, one group of 335 advanced ccRCC patients received pembrolizumab + lenvatinib combination therapy and the other group of 357 advanced ccRCC patients received sunitinib therapy alone. The results showed a PFS of 23.9 months vs. 9.2 months and an OS of (HR, 0.66; 95% CI, 0.49 to 0.88; *p* = 0.005) in the pembrolizumab + lenvatinib combination therapy group versus sunitinib therapy group, respectively. The above-mentioned results suggest that pembrolizumab + lenvatinib combination therapy as a first-line treatment of advanced ccRCC patients is superior to sunitinib therapy alone. However, the incidence of treatment-related Grade 3 or higher AEs (e.g., hypertension, diarrhea, and elevated lipase levels) was 82.4% vs. 71.8% in the pembrolizumab + lenvatinib combination therapy group versus the sunitinib therapy group, respectively.

○The phase III CheckMate 9ER clinical trial evaluated nivolumab + cabozantinib combination therapy compared to sunitinib therapy alone. In this clinical trial, one group of 323 advanced RCC patients received nivolumab + cabozantinib combination therapy and the other group of 328 advanced RCC patients received sunitinib therapy alone. The results showed a PFS of 16.6 months vs. 8.3 months, an OS of 85.7% vs. 75.6%, and an ORR of 55.7% vs. 27.1% in the nivolumab + cabozantinib combination therapy group versus the sunitinib therapy group, respectively. However, the incidence of treatment-related Grade 3 or higher AEs was 82.4% vs. 71.8% in the nivolumab + cabozantinib combination therapy group versus the sunitinib therapy group, respectively. In the nivolumab + cabozantinib combination therapy group, the AEs forced 19.7% of the patients to discontinue use of at least one of the trial drugs and 5.6% of the patients to discontinue use of both trial drugs. Overall, the advanced RCC patients in the nivolumab + cabozantinib combination therapy group had a better quality of life than the patients in the sunitinib therapy group [131].

○The phase III IMmotion151 (NCT02420821) clinical trial evaluated atezolizumab + bevacizumab combination therapy compared to sunitinib therapy alone [132]. In this clinical trial, one group of 454 RCC patients (about 40% were PD-L1^+^) received atezolizumab + bevacizumab combination therapy and the other group of 461 RCC patients received sunitinib therapy alone. The results showed a PFS of 11.2 months vs. 7.7 months in the atezolizumab + bevacizumab combination therapy group versus the sunitinib therapy group, respectively. However, the incidence of treatment-related Grade 3–4 AEs was 40% vs. 54% and treatment-related all Grade AEs was 5% vs. 8% in the atezolizumab + bevacizumab combination therapy group versus the sunitinib therapy group, respectively.

■ICIs combination with cytokine therapy

○The use of high dose IL-2 in the treatment of advanced ccRCC has been limited due to the severe toxicity of IL-2 [133]. However, in the era of ICIs, the use of IL-2 has been reconsidered. In order to reduce IL-2 toxicity and to improve IL-2 anti-tumor activity, nemvaleukin alfa (nemvaleukin; ALKS 4230) was constructed. Nemvaleukin alfa is a fusion protein of circularly permuted IL-2 to the extracellular domain of IL-2 receptor α (IL-2Rα). Nemvaleukin alfa mimics an intermediate-affinity rather than a high-affinity IL-2R which results in an IL-2 fusion protein that preferentially stimulates effector T cells rather than Tregs cells [134]. The phase I/II ARTISTRY-1 (NCT02799095) clinical trial evaluated pembrolizumab + nemvaleukin alfa combination therapy as a second-line treatment for solid tumors (including RCC). The preliminary results showed an ORR of 16.1% and a disease control rate of 59.9% in a solid tumor cohort [135].

○Bempegaldesleukin (NKTR-214) is a pegylated form of the cytokine IL-2 which preferentially binds to IL-2Rβ over IL-2Rα. The phase I PIVOT-02 (NCT02983045) clinical trial evaluated nivolumab + NKTR-214 combination therapy as a first-line treatment in 14 advanced RCC patients and as a second-line treatment in 8 patients. The results showed an ORR of 71.4% (first-line treatment) and 28.6% (second-line treatment). However, the incidence of treatment-related Grade 3–4 AEs was 21.1% in the patients. A Phase III randomized clinical trial (PIVOT-09, NCT03729245) investigated the efficacy of bempegaldesleukin in combination with nivolumab compared to sunitinib or cabozantinib in patients with previously untreated advanced ccRCC. The findings indicated an ORR of 23.0% for the combination therapy versus 30.6% for the TKI treatments, with a median OS of 29.0 months. Notably, the adverse reactions associated with the combination of bempegaldesleukin and nivolumab were predominantly pyrexia (32.6% compared to 2.0%) and pruritus (31.3% compared to 8.8%). Furthermore, the incidence of grade 3/4 TRAEs was lower in the group receiving bempegaldesleukin plus nivolumab (25.8%) compared to those receiving TKI therapy (56.5%) [136].

○Pegilodecakin is a pegylated recombinant human IL-10. The phase 1/1b IVY (NCT02009449) clinical trial evaluated pembrolizumab + pegilodecakin combination therapy compared to nivolumab + pegilodecakin combination therapy against several solid tumors [137]. In this clinical trial, one group of 53 patients received pembrolizumab + pegilodecakin combination therapy and the other group of 58 patients received nivolumab + pegilodecakin combination therapy. The results showed that 93% of patients (103 out of 111) had at least one treatment-related AE and 66% of patients had Grade 3–4 AEs. In a clinical trial involving 38 patients with advanced RCC, the ORR was found to be 40% among the 35 patients who were assessed for response, and 44% among the 27 patients who had received prior treatment [137].

The use of pegilodecakin as a monotherapy or a combination therapy with ICIs in heavily pre-treated advanced RCC patients has been reported [138]. In this study, one group of 24 patients received pegilodecakin therapy alone, the second group of 4 patients received pegilodecakin + pazopanib combination therapy, and the third group of 38 patients received pegilodecakin + anti-PD-1 inhibitor combination therapy. The results showed an ORR of 20% vs. 33% vs. 43% and a PFS of 1.8 months vs. 3.7 months vs. 13.9 months in the 3 groups, respectively. Of note, the incidence of treatment-related AEs (e.g., anemia, thrombocytopenia and hypertriglyceridemia) were similar to the previous reports, most of which were manageable [138].

**Table 3 ijms-26-11986-t003:** Clinical trials of immunotherapy for ccRCC.

Clinical Trials	Identifiers	Patients	Arms	Results
CDR0000285624 (II) [108]	NCT00053729	61	Ipilimumab	low dose ORR: 4.8% high dose ORR: 12.5%
CheckMate 025 (III) [109,110]	NCT01668784	803	Nivolumab vs. Everolimus	ORR: 23% vs. 4% OS: 25.8 mo vs. 19.7 mo
KEYNOTE- 427 (II) [112]	NCT02853344	110	Pembrolizumab	ORR:36.4% PFS:7.1 mo
IMmotion 150 (II) [114]	NCT01984242	305	Atezolizumab vs. Sunitinib	ORR:25% vs. 29% PFS:7.8 mo vs. 5.5 mo
CheckMate-214 (III) [122,124]	NCT02231749	1096	Nivolumab + Ipilimumab vs. Sunitinib	ORR: 42% vs. 27% PFS: 11.6 mo vs. 8.4 mo OS: NR vs. 26.0 mo
JAVELIN Renal 101 (III) [113,127]	NCT02684006	886	Avelumab + Axitinib vs. Sunitinib	ORR: 55.2% vs. 25.5% PFS: 13.8 mo vs. 8.4 mo OS: 10.8 mo vs. 8.6 mo
KEYNOTE-426 (III) [128,139]	NCT02853331	861	Pembrolizumab + Axitinib vs. Sunitinib	ORR: 59.3% vs. 35.7% PFS: 15.1 mo vs. 11.1 mo
CheckMate 9ER (III) [131]	NCT03141177	651	Nivolumab + Cabozantinib vs. Sunitinib	ORR: 55.7% vs. 27.1% PFS: 16.6 mo vs. 8.3 mo
KEYNOTE-581 (III) [59]	NCT02811861	1069	Pembrolizumab + Lenvatinib vs. Sunitinib	ORR: 71% vs. 36.1% PFS: 23.9 mo vs. 9.2 mo
IMmotion 151 (III) [132]	NCT02420821	915	Atezolizumab + Bevacizumab vs. Sunitinib	PD-L1^+^ patients ORR: 43% vs. 35% PFS: 11.2 mo vs. 7.7 mo ITT population ORR: 37% vs. 33% PFS: 11.2 mo vs. 8.4 mo
ALKS 4230 (I/II) [134,135]	NCT02799095	243	Pembrolizumab + Nemvaleukin alfa vs. Nemvaleukin alfa	ORR: 16.1% (combination) ORR:18.2% (monotherapy)
PIVOT-02 (I) [140]	NCT02983045	22	Nivolumab + NKTR-214 (1st line and 2nd line)	ORR: 71.4 (1st line) % and 28.6% (2nd line)
IVY(I) [137,138]	NCT02009449	62	Pembrolizumab/Nivolumab + Pegilodecakin vs. pegilodecakin	ORR: 43% vs. 20% PFS: 13.9 mo vs. 1.8 mo

ORR: objective response rate; PFS: progression-free survival; OS: Overall survival; NR: Not reached; ITT: Intention-to-treat.

## 4. Conclusions

As immunotherapy plays an increasingly important role in the treatment of ad-vanced ccRCC, the development of new ICIs and ICIs-based combination therapies provide many novel possibilities for ccRCC treatment in the future. At present, the first-line treatment for ccRCC patients with moderate to poor prognosis is mainly dual immunotherapy or ICI + VEGF combination therapy for all risk groups. Although these therapies improve the overall survival (OS), not all patients experience long-term remission. Ongoing clinical trials may continue to provide more therapy possibilities for advanced ccRCC patients. In addition, the discovery of better prognostic biomarkers can assist in determining which of the many therapies may be more beneficial to patient outcomes. However, the effectiveness and applicability of these new therapies and biomarkers will need to be explored in subsequent preclinical and clinical studies.

## Figures and Tables

**Figure 1 ijms-26-11986-f001:**
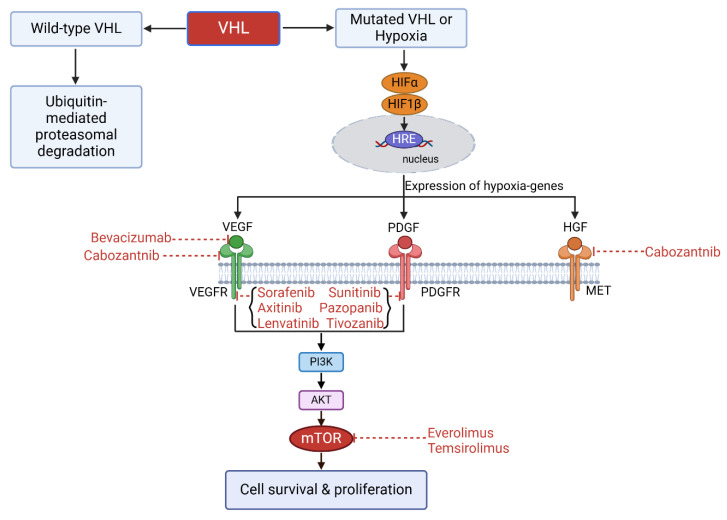
Illustration of small molecular inhibitors targeting associated molecules involved in the development of ccRCC due to VHL mutation.

**Figure 2 ijms-26-11986-f002:**
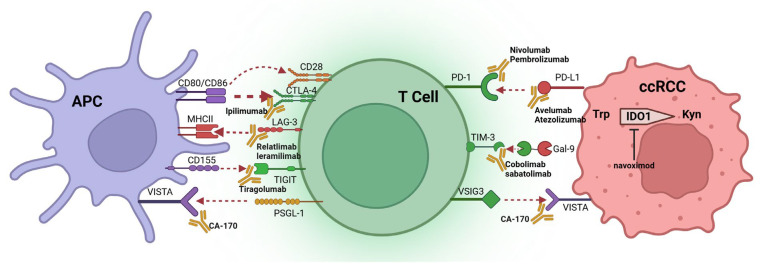
Depiction of immune checkpoints and their associated partners, along with antibodies designed to inhibit the interactions between immune checkpoints and their respective partners.

## Data Availability

No new data were created or analyzed in this study. Data sharing is not applicable to this article.

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
