# Peer review of "Immune Landscape and Application of Immune Checkpoint Inhibitors in Clear Cell Renal Cell Carcinoma"

_ijms, 2025, doi:10.3390/ijms262411986_

Round 1

Reviewer 1 Report

Comments and Suggestions for Authors

Please find my comments and suggestions attached. 

Author Response

The authors provided a comprehensive overview of ccRCC, including its pathology, gene mutations, epigenetic modifications, and the immune landscape. The authors also reviewed the development of targeted therapy and immunotherapy for ccRCC. The review is informative, while several revisions are required before publication.

  1. Line 31: The 2020 data are outdated and more recent statistics should be cited.

Particularly, there have been new drugs approved for kidney cancer after 2020, which might change the death rate. Please research and update.

We appreciate your insightful suggestions. The statistics and references have been updated accordingly.

  1. Line 132: Was the citation for “Sumeyye et al” provided in the Reference list?

Thank you for your valuable suggestions. We have already incorporated the relevant references.

  1. Line 143: The corresponding reference for “Borcherding et al” does not seem to be mentioned in the text? Please ensure the citation is properly formatted.

Thank you for your valuable suggestions. We have already incorporated the relevant references.

  1. The authors could consider moving the paragraph starting at line 181 to section “2.2 Genomic/Molecular characterization of ccRCC” and add “epigenomic” into the subtitle.

We appreciate your insightful suggestions and concur that incorporating epigenetic modifications would undoubtedly improve the quality of the manuscript. However, the purpose of this paragraph is to illustrate the role of epigenetic regulation within the tumor microenvironment, particularly concerning immune cell infiltration, rather than focusing on epigenetic regulation in the development of ccRCC.

  1. Table 1 and Figure 1 already present the targeted therapies clearly and effectively. Thus, the long text (Line 195 – 239) seems redundant and distracts from the main narrative. Please consider integrating any essential details directly into Table 1 and remove or substantially condensing the descriptive text.

We appreciate your insightful suggestions and fully concur with your recommendations. We have taken these considerations into account during the preparation of the manuscript. The rationale behind the current manuscript structure is based on the recognition that readers have diverse reading preferences. For some individuals, the textual content is more accessible and comprehensible than tabular data. Consequently, the text serves as a detailed descriptive narrative, while the tables function as concise summaries of the information presented in the text.

  1. For Table2, I suggest restructuring it to highlight the approved immunotherapies for ccRCC, along with their targets and mechanisms of action. It can be parallel with Table3 which only focuses on clinical trials. This way it avoids repetition of immune checkpoint biology that has been covered in the main text.

We appreciate your suggestions and fully concur with your recommendations. One rationale for maintaining the manuscript's current layout is that the text provides a detailed descriptive narrative, whereas the tables offer concise summaries of the information presented within the text. Furthermore, ongoing clinical trials of immunotherapy for ccRCC frequently involve combination treatments with tyrosine kinase inhibitors (TKIs), rather than immune checkpoint inhibitors (ICIs) alone. Consequently, we have decided to omit the proposed table summarizing immunotherapy in ongoing clinical trials.

  1. Line 381: “Ipilimumab + nivolumab treatment is the only approved dual combination” is not accurate. I believe avelumab + axitinib has already been approved for advanced RCC treatment. Please confirm and update the text and Tables.

We appreciate your suggestions. There appears to have been a misunderstanding regarding the phrasing of certain sentences. The intended meaning is that the ipilimumab +nivolumab regimen is the sole combination therapy comprising two immunotherapy agents. We have revised the manuscript accordingly to clarify this point. Additionally, the combination of avelumab+axitinib is classified as an ICI plus TKI therapy, as noted in Line 408.

Reviewer 2 Report

Comments and Suggestions for Authors

In the following work, Yanhe An and Na Luo review the pathological, genomic, and molecular characteristics of ccRCC, with a particular focus on its immune attributes. They also address the clinical implications of targeted therapies and immunotherapies, whether administered as monotherapy or in combination with traditional or novel agents, while evaluating the results of relevant clinical trials.

Major points

  1. In renal cancer, conventional radiotherapy has proven largely ineffective. However, the use of high doses of radiation in a few fractions, known as stereotactic ablative radiotherapy (SAR), has demonstrated a high rate of tumor growth control in both primary and metastatic renal cancer. The authors should add this point to their review, since in lines 40-41 they state: "Moreover, RCC demonstrates an insensitivity to radiation and chemotherapy."

  1. Line 68 onwards: authors should mention KIM-1 and relate changes in the expression of this protein as a possible biomarker of kidney cancer (there is a lot of literature that deals with it).

Author Response

  1. In renal cancer, conventional radiotherapy has proven largely ineffective. However, the use of high doses of radiation in a few fractions, known as stereotactic ablative radiotherapy (SAR), has demonstrated a high rate of tumor growth control in both primary and metastatic renal cancer. The authors should add this point to their review, since in lines 40-41 they state: "Moreover, RCC demonstrates an insensitivity to radiation and chemotherapy."

Thank you for your suggestions. We have already included descriptions of SAR treatment for ccRCC in the manuscript.

  1. Line 68 onwards: authors should mention KIM-1 and relate changes in the expression of this protein as a possible biomarker of kidney cancer (there is a lot of literature that deals with it).

Thank you for your suggestions. We have already included descriptions of KIM-1 and ccRCC in the manuscript.

Round 2

Reviewer 2 Report

Comments and Suggestions for Authors

The authors have responded to each of the suggestions and have modified the manuscript when required.